# PivotMesh: Generic 3D Mesh Generation via Pivot Vertices Guidance

**Haohan Weng**[1][*]   **Yikai Wang**[2][†]   **Tong Zhang**[1]   **C. L. Philip Chen**[1]   **Jun Zhu**[23]
[1]South China University of Technology   [2]Tsinghua University   [3]ShengShu

## Abstract

Generating compact and sharply detailed 3D meshes poses a significant challenge for current 3D generative models. Different from extracting dense meshes from neural representation, some recent works try to model the native mesh distribution (i.e., a set of triangles), which generates more compact results as humans crafted. However, due to the complexity and variety of mesh topology, most of these methods are typically limited to generating meshes with simple geometry. In this paper, we introduce a generic and scalable mesh generation framework PivotMesh, which makes an initial attempt to extend the native mesh generation to large-scale datasets. We employ a transformer-based autoencoder to encode meshes into discrete tokens and decode them from face level to vertex level hierarchically. Subsequently, to model the complex typology, our model first learns to generate pivot vertices as coarse mesh representation and then generate the complete mesh tokens with the same auto-regressive Transformer. This reduces the difficulty compared with directly modeling the mesh distribution and further improves the model controllability. PivotMesh demonstrates its versatility by effectively learning from both small datasets like Shapenet, and large-scale datasets like Objaverse and Objaverse-xl. Extensive experiments indicate that PivotMesh can generate compact and sharp 3D meshes across various categories, highlighting its great potential for native mesh modeling. Project Page: https://whaohan.github.io/pivotmesh

## 1 Introduction

The field of 3D generation has witnessed remarkable advancements in recent years (Poole et al., 2023; Hong et al., 2023; Xu et al., 2024a). Meshes, the predominant representation for 3D geometry, are widely adopted across various applications from video games and movies to architectural modeling. Despite the promising performance of current methods, they mostly rely on neural 3D representation like triplanes (Hong et al., 2023; Li et al., 2023) and FlexiCubes (Xu et al., 2024a). Post-processed meshes extracted from these representations tend to be dense and over-smoothed, which are unfriendly for modern rendering pipelines as shown in Figure 1 (bottom). In contrast, meshes crafted by humans are typically more compact with fewer faces, reusing geometric primitives to efficiently represent real-world objects.

To avoid extracting dense meshes through post-processing, another promising direction is emerging that focuses on explicitly modeling the mesh distribution (i.e., native mesh generation). This line of works (Nash et al., 2020; Siddiqui et al., 2023; Alliegro et al., 2023) generates meshes by predicting the 3D coordinates of faces, thus producing compact meshes as humans. However, due to the complexity and variety of topological structures in meshes, most of these methods are typically confined to generating meshes with simple topology, hindering the generalizability across complex objects. Therefore, *it still remains a challenge to establish a generic generative model for native mesh generation at scale.*

In this paper, we propose PivotMesh, a generic and scalable framework to extend mesh generation to large-scale datasets across various categories. PivotMesh consists of two parts: a mesh autoencoder and a pivot-guided mesh generator. First, the autoencoder is based on the Transformer to encode

---

[*]Work done during the internship at ShengShu    [†]Corresponding author

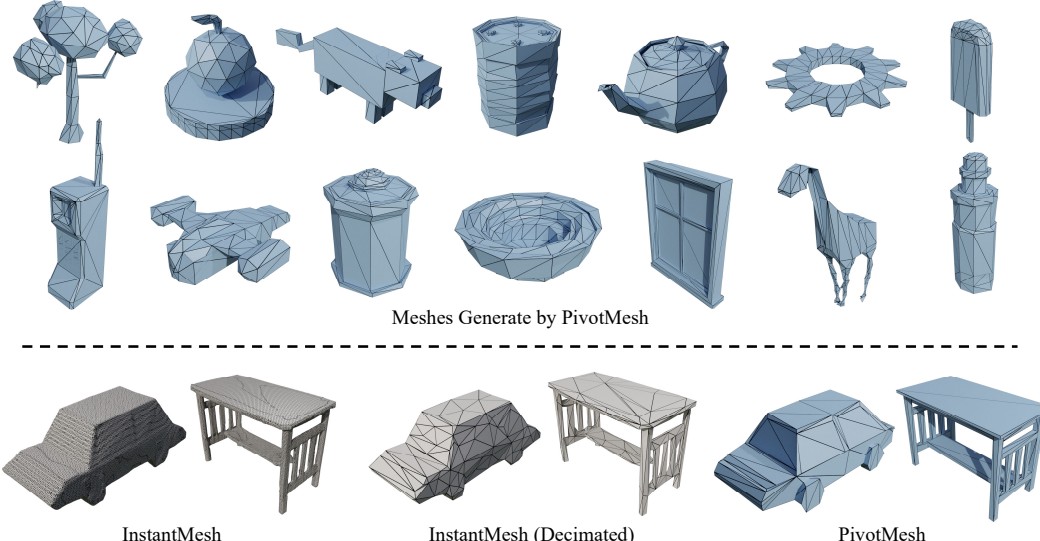

Figure 1: Different from 3D generation methods based on neural representations like InstantMesh (Xu et al., 2024a), our methods can generate compact and sharp meshes with much fewer faces when producing similar shapes.

meshes into discrete tokens. We also adopt a two-stage decoding strategy to decode mesh tokens from face level to vertex level hierarchically, which further improves the reconstruction performance and mesh surface continuity. Second, we employ an auto-regressive Transformer to learn the joint distribution of pivot vertices and complete mesh tokens, where the pivot vertices serve as the coarse representation to guide the following mesh generation. Specifically, pivot vertices are selected based on vertex degree and dropped randomly to prevent overfitting. As shown in Figure 1 (top), once the model is trained, it can produce meshes from scratch, starting with the generation of pivot vertices followed by the complete mesh token sequence. Furthermore, it can perform conditional generation given the pivot vertices from the reference mesh and supports downstream applications.

PivotMesh is designed to be scalable and extensible. We initially evaluate its effectiveness on small dataset ShapeNet (Chang et al., 2015) as previous settings (Siddiqui et al., 2023). Next, we carefully curate and train our model on the existing largest 3D datasets Objavese (Deitke et al., 2023) and Objaverse-xl (Deitke et al., 2024). By leveraging large datasets, our model can generate generic meshes across various categories to accelerate the mesh creation process. Both the qualitative and quantitative experiments show that the proposed PivotMesh beats previous mesh generation methods like PolyGen (Nash et al., 2020) and MeshGPT (Siddiqui et al., 2023) by a large margin.

The contributions of this paper can be summarized as follows:

- We propose a generic and scalable mesh generation framework PivotMesh, which makes an initial attempt to extend the native mesh generation to large-scale datasets.
- We present a Transformer-based autoencoder to preserve the geometry details and surface continuity in meshes by efficiently decoding from face level to vertex level hierarchically.
- We introduce pivot vertices guidance for complex mesh geometry modeling, which serves as the coarse representation to guide the complete mesh generation in a coarse-to-fine manner.
- PivotMesh achieves promising performance in various applications like mesh generation, variation, and refinement, accelerating the mesh creation process.

## 2 RELATED WORKS

**Neural 3D Shape Generation.** Most previous attempts learn 3D shape with various representations, e.g., SDF grids (Cheng et al., 2023; Chou et al., 2023; Shim et al., 2023; Zheng et al., 2023) and neural fields (Gupta et al., 2023; Jun & Nichol, 2023; Müller et al., 2023; Wang et al., 2023a; Zhang

Table 1: **Difference between MeshGPT and two concurrent works with PivotMesh.** Our main contribution mainly falls on two aspects. First, our autoencoder yields great reconstruction performance with a shorter sequence length. Second, our model can produce a more complex typology with the pivot guidance. $n$ is the number of faces and $v$ is the number of vertices.

| Difference | MeshGPT | MeshXL | MeshAnything | PivotMesh |
|---|---|---|---|---|
| AE Architecture | GNN-CNN | N/A | Transformer | Transformer |
| AE Decoding | Face-level | N/A | Face-level | Face-level & Vertex-level |
| Sequence Type | Latent | Coordinates | Latent | Pivot-guided Latent |
| Generation Formulation | Direct | Direct | Direct | Coarse-to-fine |
| Sequence Length | $6n$ | $9n$ | $9n$ | $0.1v + 6n$ |
| Compression Ratio ($\downarrow$) | 66.7% | 100% | 100% | 68.9% |

et al., 2023; Liu et al., 2023b; Lyu et al., 2023). To improve the generalization ability, researchers start to leverage pretrained 2D diffusion models (Rombach et al., 2022; Saharia et al., 2022; Liu et al., 2023a) with score distillation loss (Poole et al., 2023; Lin et al., 2023; Wang et al., 2023b) in a per-shape optimization manner. Multi-view diffusion models (Shi et al., 2023b; Weng et al., 2023; Zheng & Vedaldi, 2023; Shi et al., 2023a; Chen et al., 2024d; Voleti et al., 2024) are used to further enhance the quality and alleviate the Janus problem. Recently, Large Reconstruction Models (LRM) (Hong et al., 2023; Li et al., 2023; Xu et al., 2023; Wang et al., 2024; Xu et al., 2024b; Tang et al., 2024a; Xu et al., 2024a) train the Transformer backbone on large scale dataset (Deitke et al., 2023) to effectively generates generic neural 3D representation and shows the great performance of scaling. However, these neural 3D shape generation methods require post conversion (Lorensen & Cline, 1998; Shen et al., 2021) for downstream applications, which is non-trivial and easy to produce dense and over-smooth meshes.

**Native Mesh Generation.** Compared with the well-developed generative models of neural shape representations, the generation of the mesh remains under-explored. Some pioneering works try to tackle this problem by formulating the mesh representation as surface patches (Groueix et al., 2018), deformed ellipsoid (Wang et al., 2018), mesh graph (Dai & Nießner, 2019) and binary space partitioning (Chen et al., 2020). PolyGen (Nash et al., 2020) uses two separated auto-regressive Transformers to learn vertex and face distribution respectively. Polydiff (Alliegro et al., 2023) learns the triangle soups of mesh with a diffusion model. MeshGPT (Siddiqui et al., 2023) is most relevant to our work, which first tokenizes the mesh representation with a GNN-based encoder and learns the mesh tokens with a GPT-style Transformer. Despite its promising results on small datasets, it is non-trivial to extend MeshGPT to the large-scale dataset. Our research, along with several concurrent works (Chen et al., 2024a;b;c; Tang et al., 2024b) as shown in Table 1, is designed to build a generic generative model for native mesh generation within large-scale datasets.

## 3 METHOD

In this section, we will introduce the details of the proposed PivotMesh as shown in Figure 2. The challenges to scale up native mesh generation are analyzed in Section 3.1. Meshes formulated as triangle face sequences are first encoded into discrete tokens by the proposed mesh autoencoder (Section 3.2). Then, we use an auto-regressive Transformer to learn the joint distribution of pivot vertices and mesh tokens (Section 3.3).

### 3.1 CHALLENGES FOR NATIVE MESH GENERATION ON LARGE DATASETS

There are two main challenges for scaling up native mesh generation to large datasets.

**Mesh Reconstruction.** It is challenging to tokenize meshes due to its high requirement on reconstruction accuracy to preserve the mesh surface continuity. Previous works like MeshGPT (Siddiqui et al., 2023) formulate meshes as face graphs for reconstruction, which only focuses on face-level relationships and neglects the connection and interaction among vertices. Furthermore, the limited network capability of autoencoder (i.e., GNN and CNN) also hinders its scalability on large-scale

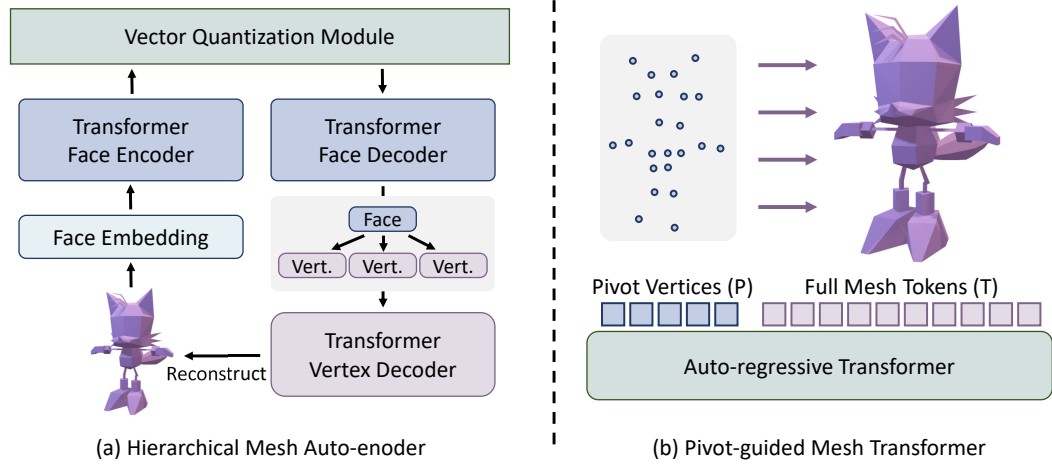

Figure 2: **The overall method of PivotMesh.** (a) Triangle mesh sequences are tokenized into mesh tokens and hierarchically decoded from face level to vertex level via our mesh autoencoder. (b) The auto-regressive Transformer first learns to generate pivot vertices as coarse mesh representation and then generates the complete mesh tokens in a coarse-to-fine manner.

datasets. To this end, we propose a Transformer-based autoencoder to preserve the geometry details and surface continuity by decoding from face level to vertex level hierarchically.

**Complex Typology Modeling.** Due to the complexity and variety of mesh topology, directly modeling the mesh sequence on large-scale datasets makes it easy to produce trivial meshes with simple geometry (e.g., cubes). To model complex mesh typology, a natural solution is to first generate a coarse representation and then the full mesh sequence. For this purpose, we define pivot vertices (the sequence of high-degree vertices) as the coarse representation of meshes. With the guidance of pivot vertices, our model is capable of generating complex mesh geometry in a coarse-to-fine manner.

### 3.2 ENCODE MESHES INTO DISCRETE TOKENS

A triangle mesh $\mathcal{M}$ with $n$ faces can be formulated as the following sequence:
$$\mathcal{M} := (f_1, f_2, ..., f_n) = (v_{11}, v_{12}, v_{13}, v_{21}, v_{22}, v_{23}, ..., v_{n1}, v_{n2}, v_{n3}), \tag{1}$$
where each face $f_i$ consists of 3 vertices and each vertex $v_i$ contains 3D coordinates discretized with a 7-bit uniform quantization. To effectively learn the mesh distribution, we first tokenize the sequence into discrete tokens with the proposed transformer-based autoencoder.

**Attention-based Tokenizer.** Different from MeshGPT equipped with a GNN-CNN-based autoencoder, we employ a Transformer-based architecture as the backbone for the encoder, capturing the long-range relationship between faces. Furthermore, we replace the vanilla positional encoding in the Transformer with a single-layer GNN to capture the local topology of meshes. This preserves the permutation invariance of faces with higher scalability, yielding more effective and robust token representation for meshes.

**Hierarchical Decoding.** To further improve the reconstruction performance and mesh surface contiguity, we design a hierarchical decoder from face level to vertex level. The face embedding $F_i'$ from the vector quantization module is first passed to a face-level decoder. Then, the decoded face embedding is converted to vertex embedding $V_i'$ by a simple MLP. The vertex embedding is then decoded by a vertex-level decoder, whose architecture is similar to the face decoder except that its input sequence is 3 times longer.
$$\begin{aligned} F_i' &= \text{FaceDec}(F_i'), \\ V_i' &= \text{VertexDec}(\text{MLP}_{n \to 3n}(F_i')), \end{aligned} \tag{2}$$
The final decoded vertex embedding $V_i$ is then converted to the quantized 3D coordinate logits $\in (1, 2, ..., 2^7)$ for each axis ($x$, $y$ and $z$) and computes the cross entropy with the input mesh

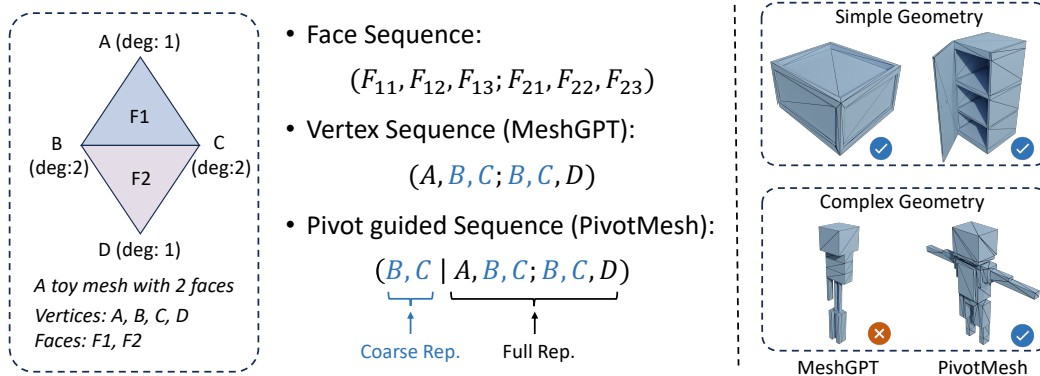

Figure 3: **Illustration for pivot guided mesh sequence.** It is hard to directly generate vertex sequence with geometry details (MeshGPT formulation) on large scale datasets. In our paper, we found that some vertices (with high degrees) repeatedly occur in vertex sequence thus we define them as pivot vertices for coarse mesh representation. By first generating pivot vertices and then the full meshes, our model is capable of producing more complex geometry with higher quality.

sequence. Such hierarchical architecture allows the connection and interaction among both face and vertex level, thus improving the reconstruction accuracy and surface continuity.

## 3.3 GUIDE MESH GENERATION WITH PIVOT VERTICES

To model complex mesh typology, it is natural to first generate a coarse representation and then the full mesh sequences. However, it is non-trivial to find such a coarse representation to preserve typology information with short length. As shown in Figure 3, we found that some vertices repeatedly occur in the mesh sequence (since such vertices connect multiple faces), therefore they are highly informative. Furthermore, these vertices are with high degrees thus preserving more geometry details. To this end, we define these vertices as pivot vertices, and propose to first generate them as the coarse representation for full mesh sequence.

**Degree-based Pivot Vertices Selection.** First, we need to select the most frequently occurring vertices as the pivot vertices. Specifically, a mesh can be regarded as a graph, where each vertex $v_i$ represents a node and the connection between vertices represents the edges. Then, we compute the vertex degree $deg(v_i)$ in mesh graphs and select the pivot vertices set $P$ with the top-degree vertices. The size of the pivot vertices set is proportional to the number of vertices with a fixed ratio $\eta_{select}$. Furthermore, to prevent overfitting in pivot-to-mesh modeling, we randomly drop some pivot vertices with the ratio $\eta_{drop}$ of all vertices for each training iteration. In our experiments, the select ratio $\eta_{select} = 15\%$ and the dropping ratio $\eta_{drop} = 5\%$, yielding the final pivot vertex ratio $\eta = 10\%$. The benefits of our pivot selecting strategy fall into two aspects. First, it leverages frequently occurring vertices (with higher degree), enabling the Transformer to utilize these as conditional tokens mesh sequence generation efficiently. Second, it tends to preserve intricate mesh details, as regions with finer geometry typically necessitate more faces and thus larger vertex degrees.

**Coarse-to-fine Mesh Modeling.** As shown in Figure 3, we employ a standard auto-regressive Transformer decoder to learn the joint distribution of the pivot vertex tokens $p_i \in P$ and the complete mesh tokens $t_i \in T$. A learnable start and end token are used to identify the beginning and end of the token sequence, while a pad token is used to separate the pivot vertex tokens and mesh tokens. The order of both pivot vertices tokens and full mesh tokens is sorted by $z$-$y$-$x$ coordinates from lowest to highest. Different from the Transformer in Section 3.2, we add absolute positional encoding here to indicate the position in the token sequence. The token sequences are modeled with a Transformer with parameter $\theta$ by maximizing the log probability:

$$\prod_{i=1}^{|T|} p(t_i | t_{1:i-1}, P; \theta) \prod_{j=1}^{|P|} p(p_j | p_{j:j-1}; \theta), \tag{3}$$

PolyGen          MeshGPT          PivotMesh

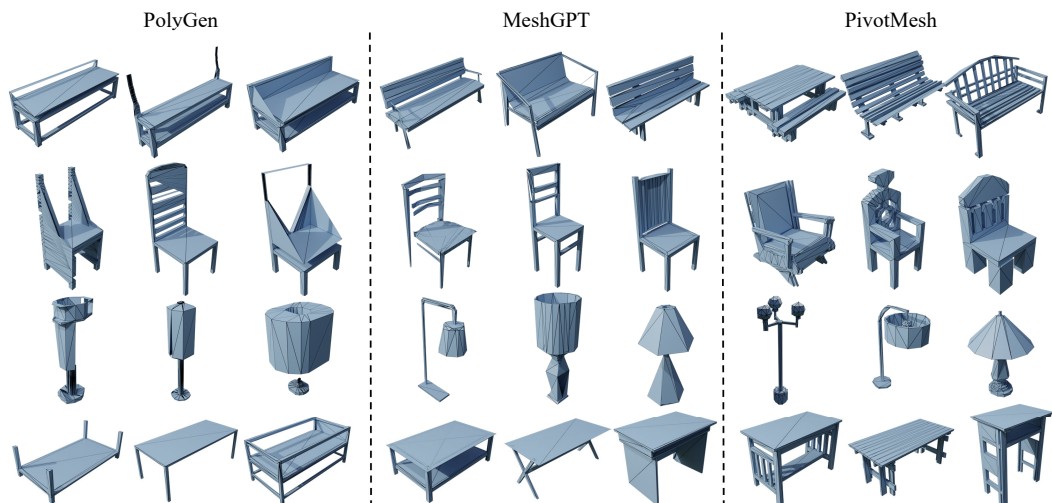

Figure 4: **Qualitative comparison of unconditional generation on ShapeNet.** Each line represents a subset of ShapeNet (bench, chair, lamp, table).

With such formulation, the auto-regressive Transformer first learns to generate pivot vertex tokens $P$ as coarse mesh representation and then generates the complete mesh tokens $T$ in a coarse-to-fine manner. In such a coarse-to-fine manner, our model can effectively learn the complex mesh typology, which can be easily extended to large-scale datasets. Once the model is trained, it can produce meshes from scratch, starting with the generation of pivot vertices followed by the full mesh sequence. Furthermore, different from previous mesh generation methods (Nash et al., 2020; Siddiqui et al., 2023), our model can perform conditional generation given the pivot vertices from the reference mesh, and also supports downstream applications like mesh variation and refinement.

## 4 EXPERIMENT

### 4.1 EXPERIMENT SETTINGS

**Datasets.** Our model is trained on various classes and scales datasets, including ShapeNetV2 (Chang et al., 2015), Objaverse (Deitke et al., 2023), Objaverse-xl (Deitke et al., 2024). For ShapeNet, follow previous settings (Nash et al., 2020; Siddiqui et al., 2023), we use 4 subsets (chair, table, bench, lamp) and filter the faces less than 800 after plannar decimation (with a fixed angle tolerance $\alpha = 10°$). For Objaverse and Objaverse-xl, we apply the data curation and filter the objects whose faces are less than 500 *without decimation* to preserve the mesh quality. The final dataset size after filtering of Shapenet, Objaverse, and Objaverse-xl is around 10k, 40k, and 400k respectively. For each dataset, we split 1k samples for testing, and leave the rest as the training data. For data augmentation, we use random scaling on each axis and random shifts ($\pm 0.01$) to enhance the data diversity.

**Baselines.** We benchmark our approach against leading mesh generation methods: **PolyGen** (Nash et al., 2020), which generates polygonal meshes by first generating vertices followed by faces conditioned on the vertices with two separate Transformer; **MeshGPT** (Siddiqui et al., 2023), which tokenizes the mesh sequence into mesh tokens with a GNN-ResNet based autoencoder and learns the mesh tokens directly with an Auto-regressive Transformer. We only visually compare with **MeshXL** (Chen et al., 2024a) since we use different training data and can not directly compute the metrics.

**Metrics.** Following the evaluation settings of previous mesh generation methods (Siddiqui et al., 2023; Alliegro et al., 2023), we evaluate the reconstruction quality by two metrics, triangle accuracy and l2 distance. We use the following metrics for mesh quality assessment: Minimum Matching Distance (MMD), Coverage (COV), and 1-Nearest-Neighbor Accuracy (1-NNA). For MMD, lower is better; for COV, higher is better; for 1-NNA, 50% is the optimal. We use the Chamfer Distance (CD) to compute these metrics on 1024-dim point clouds uniformly sampled from meshes.

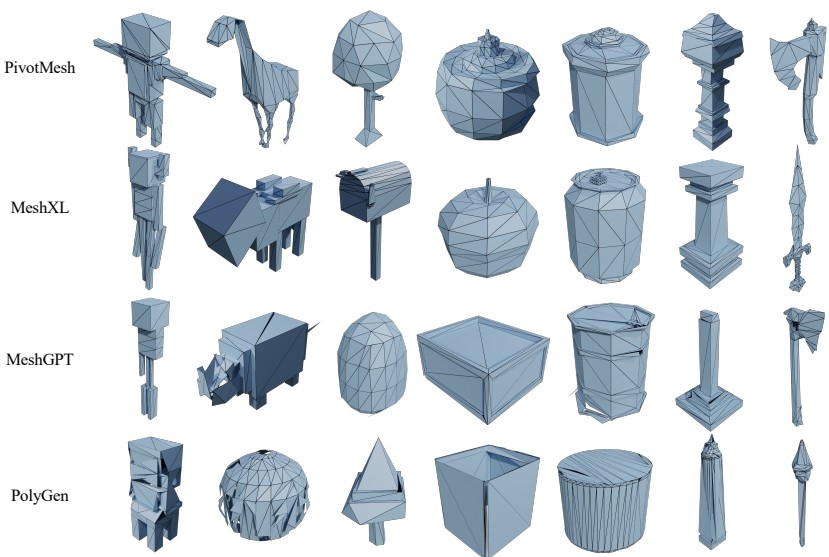

Figure 5: **Qualitative comparison of unconditional generation on Objaverse.**

Table 2: **The unconditional generation results on Shapenet dataset.** The proposed PivotMesh significantly improves the mesh quality compared with baselines by a large margin.

| Model | PolyGen | | | MeshGPT | | | PivotMesh | | |
|---|---|---|---|---|---|---|---|---|---|
| Subset | COV(%)↑ | MMD($10^{-3}$)↓ | 1-NNA(%)↓ | COV(%)↑ | MMD($10^{-3}$)↓ | 1-NNA(%)↓ | COV(%)↑ | MMD($10^{-3}$)↓ | 1-NNA(%)↓ |
| Chair | 29.47 | 13.34 | 81.45 | 41.31 | 10.30 | 61.84 | **52.89** | **9.77** | **56.71** |
| Table | 38.67 | 15.84 | 66.27 | 43.00 | 9.77 | 62.83 | **51.68** | **9.28** | **56.55** |
| Bench | 37.50 | 10.8 | 79.69 | 46.87 | 9.48 | 67.19 | **51.56** | **8.50** | **53.91** |
| Lamp | 31.76 | 33.87 | 81.76 | 45.88 | 23.43 | 57.06 | **50.58** | **22.65** | **54.71** |
| Mixed | 33.91 | 13.47 | 74.63 | 43.89 | 11.48 | 63.24 | **50.42** | **11.03** | **60.89** |

**Implementation Details.** For the autoencoder, the face encoder has 12 layers with a hidden size of 512, the face decoder has 6 layers with a hidden size of 512, and the vertex decoder has 6 layers with a hidden size of 256. For vector quantization, the number of residual quantizers $r = 2$, and the codebook is dynamically updated by exponential moving averaging with codebook size 16384 and codebook dimension 256. It is trained on an 8×A100-80GB machine for around 1 day with a batch size of 64 for each GPU. For auto-regressive Transformer, it has 24 layers with a hidden size of 1024. It is trained on an 8×A100-80GB machine for around 3 days with batch size 12 for each GPU. The temperature used for sampling is set to 0.5 to balance the quality and diversity. We use flash attention for all Transformer architecture and fp16 mixed precision to speed up the training process. We use AdamW Loshchilov & Hutter (2017) as the optimizer with $\beta_1 = 0.9$ and $\beta_2 = 0.99$ with a learning rate of $10^{-4}$ for all the experiments.

## 4.2 MESH GENERATION FROM SCRATCH

**Comparison with baselines on various-scale benchmarks.** We first evaluate the proposed PivotMesh on the commonly used benchmark Shapenet with four selected categories, chair, table, bench,

Table 3: **The unconditional generation results on the Objaverse and Objaverse-xl dataset.**

| Model | Objaverse | | | Objaverse-xl | | |
|---|---|---|---|---|---|---|
| Dataset | COV(%)↑ | MMD($10^{-3}$)↓ | 1-NNA(%)↓ | COV(%)↑ | MMD($10^{-3}$)↓ | 1-NNA(%)↓ |
| PolyGen | 23.86 | 24.01 | 84.07 | 21.79 | 22.68 | 83.40 |
| MeshGPT | 35.03 | 17.30 | 63.86 | 41.50 | 14.76 | 64.25 |
| PivotMesh | **46.48** | **16.66** | **58.55** | **45.30** | **14.33** | **57.75** |

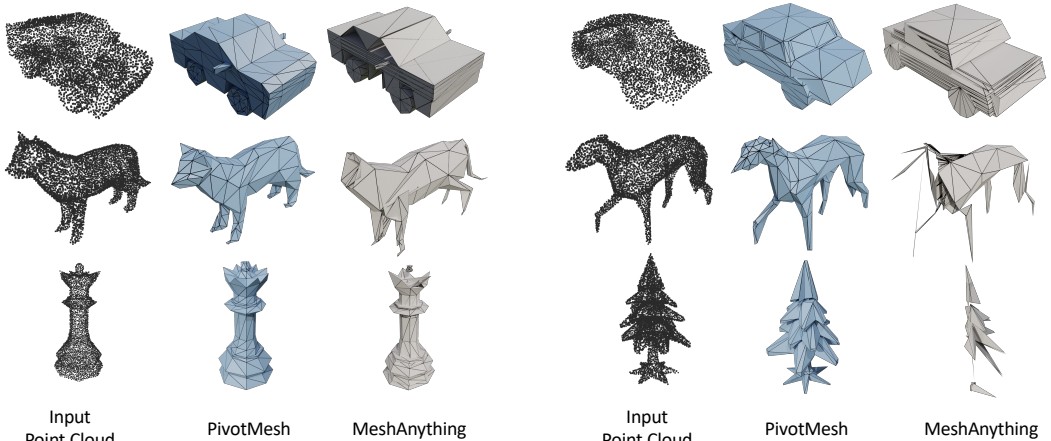

Figure 6: **Further conditioning on point clouds.** It shows that our model can produce more complex typology with the pivot vertex guidance.

Table 4: **Ablation study of autoencoder and auto-regressive Transformer on Objaverse.** (a) The proposed autoencoder reconstructs meshes with much higher accuracy compared with MeshGPT, showing the effectiveness of Transformer architecture and hierarchical decoding strategy. (b) Without the degree-based selection strategy (i.e., random selecting pivot vertices), the generation performance will greatly reduce and even worse than that without pivot vertices, showing the importance of degree-based selection strategy.

| Method | Accuracy(%)↑ | L2 Distance↓ |
|---|---|---|
| w/o Transformer (MeshGPT) | 86.89 | 10.34 |
| w/o hierarchical decode | 92.30 | 5.37 |
| Ours | **97.89** | **1.05** |

| Method | COV(%)↑ | MMD($10^{-3}$)↓ | 1-NNA(%)↓ |
|---|---|---|---|
| w/o pivot guidance | 42.76 | 16.83 | 61.38 |
| w/o degree selection | 42.48 | 17.62 | 62.90 |
| Ours | **46.48** | **16.66** | **58.55** |

(a) Ablation study on autoencoder.   (b) Ablation study on auto-regressive transformer.

and lamp. Following the previous setting (Siddiqui et al., 2023; Alliegro et al., 2023), we first pretrain our model on the mixture dataset of four selected categories and then finetune each category separately. We report the generation results both on the mixed dataset and each subset in Table 2. Furthermore, we train our model on the larger scale datasets Objaverse and Objaverse-xl and report the performance in Table 3. For all these experiments, our method can achieve state-of-the-art performance on all evaluation metrics. As shown in Figure 4 and Figure 5, our model can generate meshes with the best visual quality and geometry complexity. MeshGPT can produce complete meshes but it is trapped in simple geometry due to its network capability and the complex mesh sequence. With the hierarchical autoencoder and pivot vertices guidance, our model can produce compact meshes with sharp details and complex geometry. Besides unconditional generation, we also compare with MeshAnything in point cloud conditioning as shown in Figure 6. Our model shows the advantages of modeling complex mesh geometry.

**Shape Novelty Analysis.** To show that our model can create novel shapes instead of memorizing the training set, we conduct a shape novelty analysis similar to (Hui et al., 2022; Erkoç et al., 2023; Siddiqui et al., 2023). We generate 500 shapes and search the 3 closest neighbors from the training set measured in Chamfer Distance (CD), shown in Figure 7 (left). We also show top-1 CD distribution between the generated shapes and the training set in Figure 7 (right). The CD results show that our method not only covers shapes in the training set (low CD values) but also creates novel and realistic shapes (high CD values).

## 4.3 ABLATION STUDY

**The effectiveness of Transformer-based hierarchical autoencoder.** For mesh autoencoder, previous works like MeshGPT (Siddiqui et al., 2023) employ GNN as the encoder and CNN as the

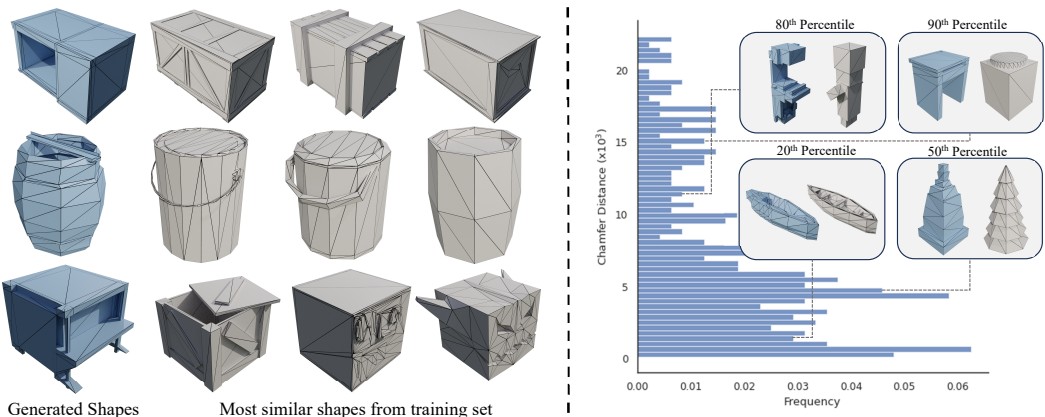

Figure 7: **Shape novelty analysis on Objaverse dataset.** We show the 3 nearest neighbors measured in Chamfer Distance (CD) for generated shapes (left). We plot the distribution of 500 generated shapes from our method and their minimum CD to the training set (right). Shapes at the 50th percentile look different from the closest train shape. It shows that our method not only covers shapes in the training set (low CD values) but also creates novel and realistic shapes (high CD values).

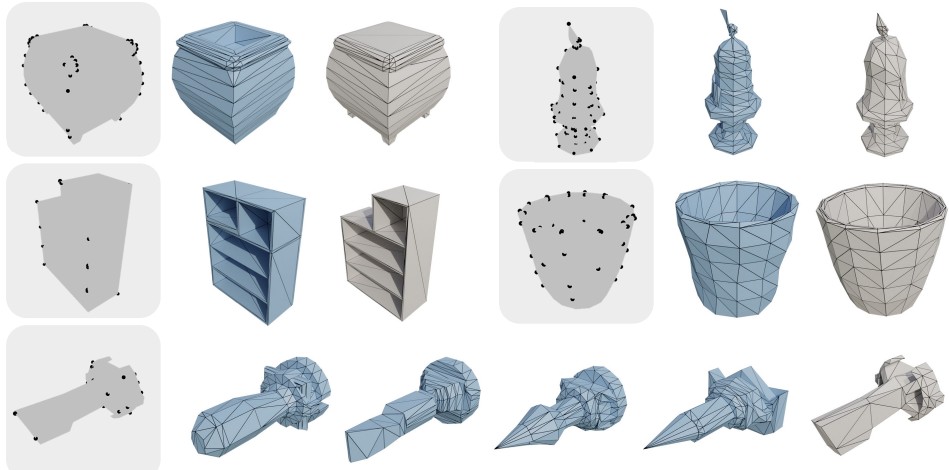

Figure 8: **Conditional generation results with pivot vertices guidance from Objaverse test set.** The generated meshes are marked in blue and the ground truth meshes are marked in gray. The diversity of generation results is also shown on the bottom line.

decoder. Table 4 (w/o Transformer) shows the reconstruction performance of such architecture, which is significantly lower than our Transformer-based autoencoder. The performance gain is mainly from the high capability and scalability of the Transformer compared with the GNNs and CNNs. Table 4 (w/o hier. decode) shows the performance of the autoencoder when removing the hierarchical decoding mechanism. It shows that such a hierarchical network design endows the decoder with better alignment in both face and vertex levels, thus improving the final reconstruction results.

**The effectiveness of pivot vertices guidance and their selection.** Table 4 (w/o pivot guidance) shows the results of directly employing an auto-regressive Transformer to model the mesh tokens from our autoencoder. Without the pivot vertex guidance, the model fails to produce meshes with complex geometry. In Table 4 (w/o degree selection), we employ random pivot vertex selection instead of the proposed degree-based strategy. Some metrics from its results are even worse than that without using pivot vertices guidance. The strong performance degradation is because the randomly selected vertices make it hard to summarize the whole geometry of the meshes and the Transformer is incapable of learning such pivot-mesh joint distribution, showing the importance of pivot vertices selection strategy.

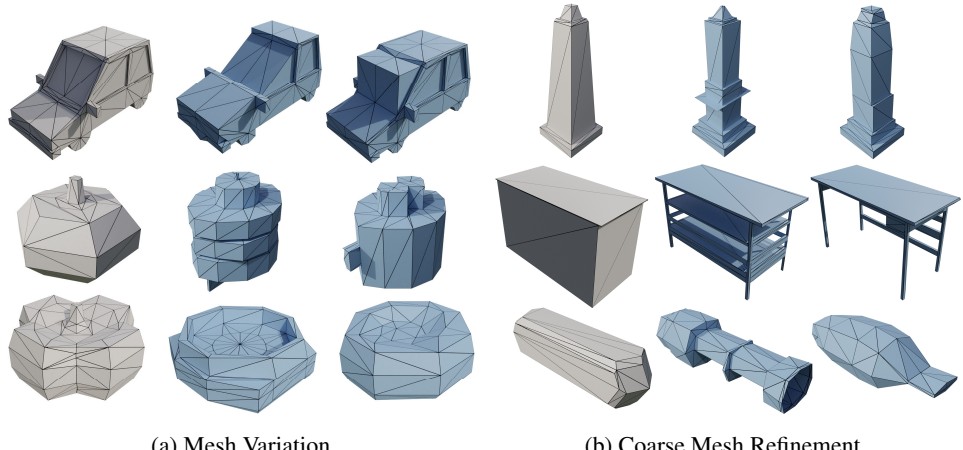

(a) Mesh Variation            (b) Coarse Mesh Refinement

Figure 9: **PivotMesh can support various downstream applications** (Reference meshes are marked in gray). (a) Mesh variation: PivotMesh generates diverse meshes similar to reference meshes but with different details. (b) Coarse Mesh Refinement: PivotMesh can refine the details for coarse meshes to accelerate the mesh creation.

### 4.4 PIVOT VERTICES GUIDANCE ANALYSIS

To trade-off between the generalization and the visual quality for pivot-guided mesh generation, we first pretrain our model in the mixture of objaverse and objaverse-xl datasets. Then, we finetune the pretrained model on the well-curated objaverse to further improve the generated mesh quality.

**Pivot-guided Mesh Generation.** Given a reference mesh, we first encode it into mesh tokens and then select the pivot vertices. Our model can generate the corresponding meshes as shown in Figure 8. It shows that our model can generate diverse and high-quality meshes while maintaining the high-level structure corresponding to the pivot vertices.

**Downstream Applications.** PivotMesh can serve as a generic mesh generative model to support various applications in Figure 9. For mesh variation, our model can generate diverse variants with the user-given meshes. For mesh refinement, it can be regarded as a special case of mesh variation, but with coarse meshes as input. Our model can refine the coarse meshes to fine meshes with detailed geometry to accelerate the mesh creation process.

### 4.5 LIMITATIONS AND FUTURE WORK

Although PivotMesh can produce compact meshes with high quality, it still has some limitations. First, the controlling ability of PivotMesh is still not enough. Since the same pivot vertices may indicate diverse meshes, our method may sometimes produce undesired geometry. This can be alleviated by adding more control conditions like images and texts and we leave it for our future work. Second, the scale of data and the number of model parameters is still limited. Due to the constrained computation resources, we only train PivotMesh on meshes less than 500 for both Objaverse and Objaverse-xl, and the number of model parameters is much less than recent advances in large language models (Touvron et al., 2023).

## 5 CONCLUSION

In this paper, we introduce PivotMesh, a scalable framework to generate generic meshes with compact and sharp geometry. By employing pivot vertices as a coarse representation to guide the mesh generation process and leveraging a Transformer-based hierarchical autoencoder, PivotMesh demonstrates its capability to generate high-quality meshes on both small and large-scale datasets. The proposed model significantly outperforms existing methods and sets a new benchmark for native mesh generation tasks, showcasing its potential for creating novel shapes and supporting downstream applications.

ACKNOWLEDGMENT

This work was funded in part by the National Natural Science Foundation of China grant under number 62222603, in part by the STI2030-Major Projects grant from the Ministry of Science and Technology of the People's Republic of China under number 2021ZD0200700, in part by Youth Fund of the National Natural Science Foundation of China (No. 62306163), and in part by China Postdoctoral Science Foundation (No. 2023M741951), in part by the Key-Area Research and Development Program of Guangdong Province under number 2023B0303030001, in part by the Program for Guangdong Introducing Innovative and Entrepreneurial Teams (2019ZT08X214), in part by the Science and Technology Program of Guangzhou under number 2024A04J6310.

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

## A ADDITIONAL RESULTS

### A.1 FAILURE CASES

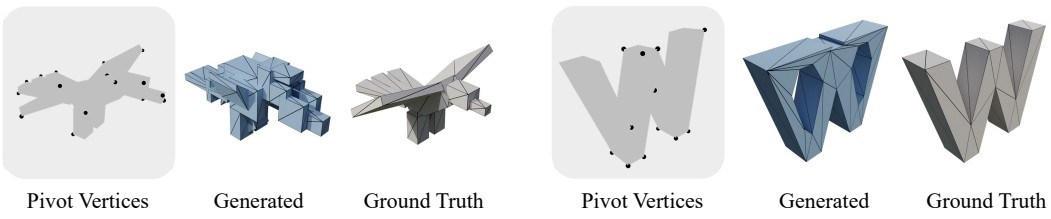

| Pivot Vertices | Generated | Ground Truth | Pivot Vertices | Generated | Ground Truth |

Figure 10: **Some failure cases for the pivot-guided mesh generation.** Since the same pivot vertices may indicate diverse meshes, our method may sometimes produce undesired geometry. This can be alleviated by adding more control conditions like images and texts.

### A.2 USER STUDY

As shown in Table 5, we conduct the user study on two dimensions Aesthetics and Complexity scores, which range from 0-5 (higher is better).

Table 5: **User study on the generated meshes.** The users are asked to score on two dimensions mesh aesthetics and complexity, which range from 1 to 5 (higher is better).

| Method | Aesthetics↑ | Complexity↑ |
|---|---|---|
| PolyGen | 2.49 | 2.35 |
| MeshGPT | 3.20 | 2.50 |
| PivotMesh | **3.60** | **3.38** |

### A.3 COMPARISON WITH INSTANTMESH

To produce similar shapes for a more straightforward visual comparison, we use the rendering images of our generated mesh as the image condition for InstantMesh to generate the following instances in Figure 1 and Figure 11.

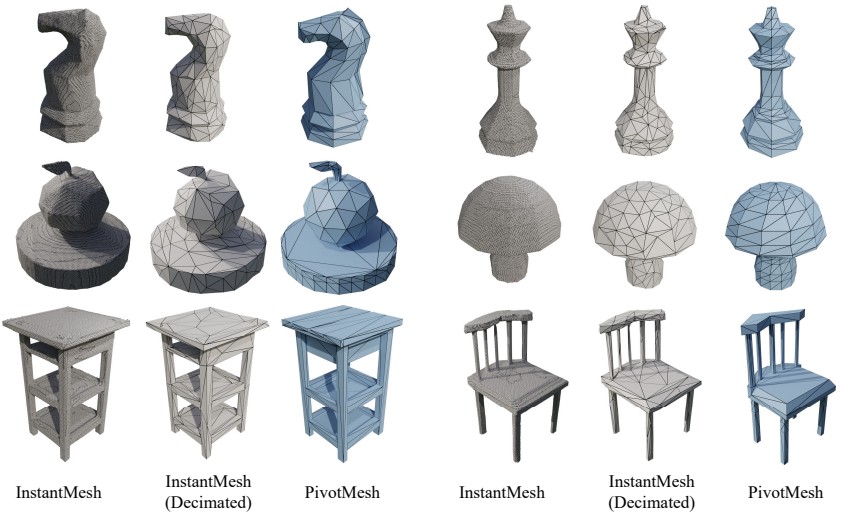

| InstantMesh | InstantMesh (Decimated) | PivotMesh | InstantMesh | InstantMesh (Decimated) | PivotMesh |

Figure 11: **Additional results for comparison with InstantMesh.**

## A.4 Unconditional Generation on ShapeNet

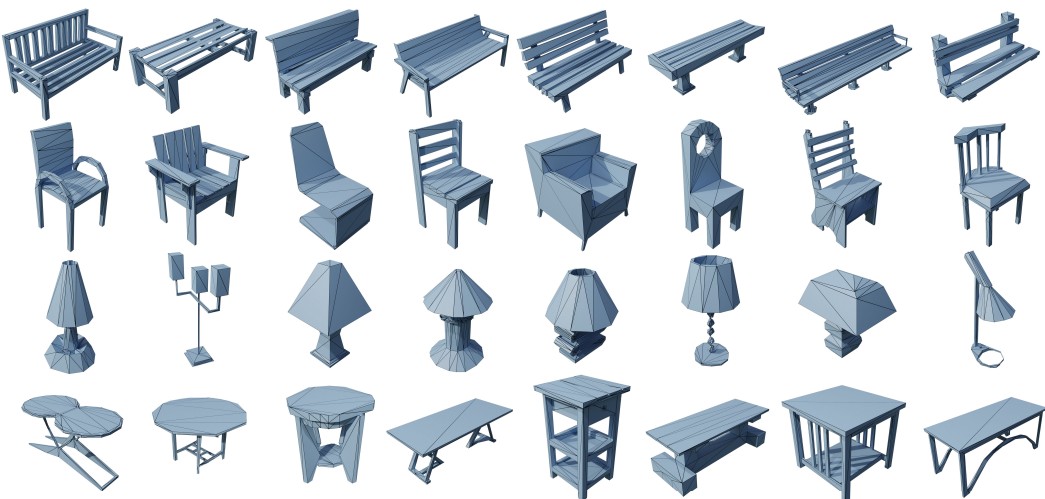

Figure 12: **Additional results for unconditional generation on Shapenet.**

## A.5 UNCONDITIONAL GENERATION ON OBJAVERSE

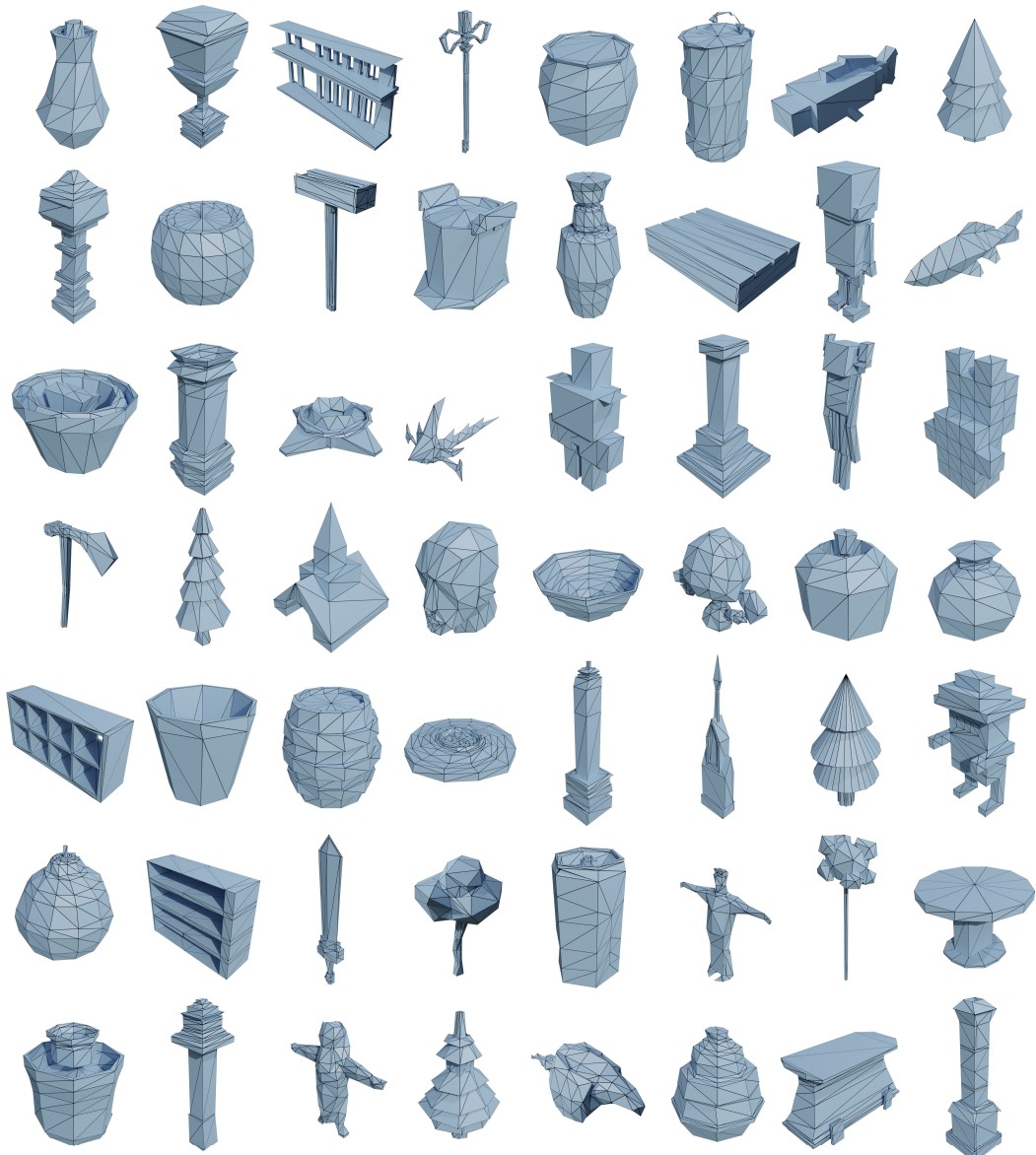

Figure 13: **Additional results for unconditional generation on Objaverse.**

## A.6 PIVOT-GUIDED MESH GENERATION

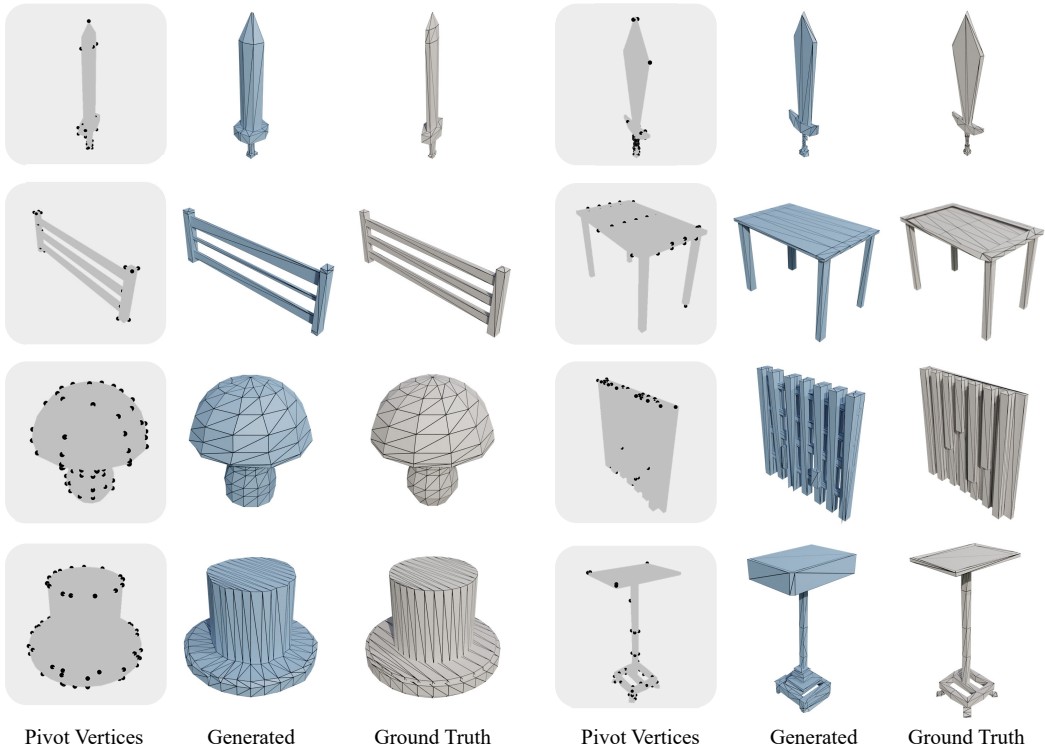

Pivot Vertices     Generated     Ground Truth     Pivot Vertices     Generated     Ground Truth

Figure 14: **Additional results for pivot-guided mesh generation on Objaverse.**

## A.7 SHAPE NOVELTY ANALYSIS

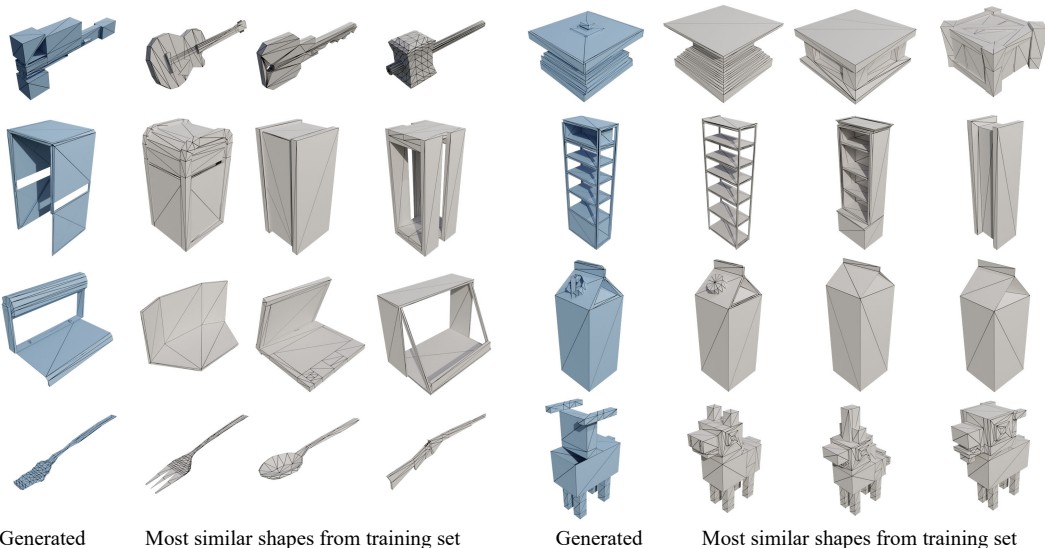

Generated     Most similar shapes from training set     Generated     Most similar shapes from training set

Figure 15: **Additional results for shape novelty analysis on Objaverse.**

## A.8    PIVOTMESH APPLICATIONS

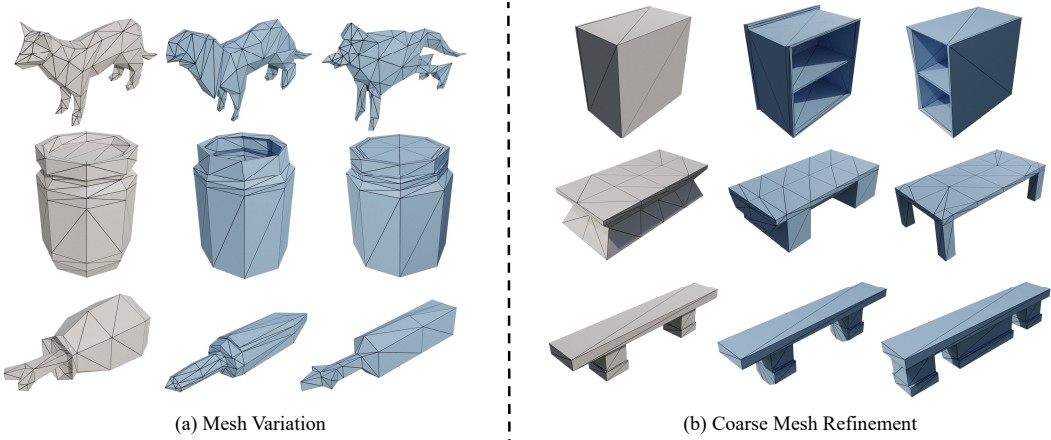

(a) Mesh Variation                    (b) Coarse Mesh Refinement

Figure 16: **Additional results for the applications of PivotMesh.**

## A.9    MESHES WITH COMPLEX TOPOLOGIES

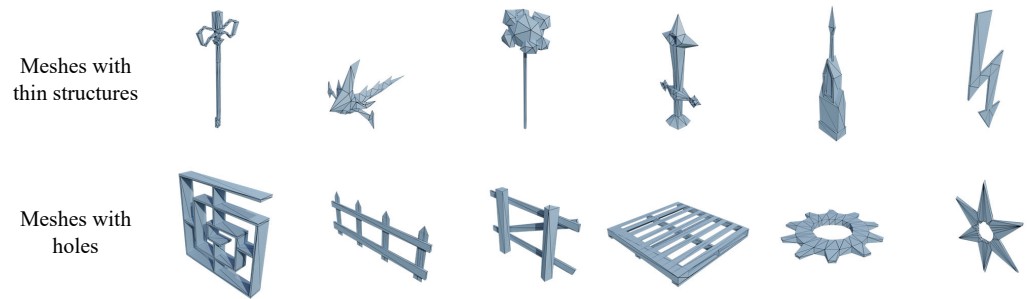

Figure 17: Generated Meshes with more complex topologies(especially for meshes with holes and thin structures.)

## A.10    COMPARISON WITH MORE METHODS

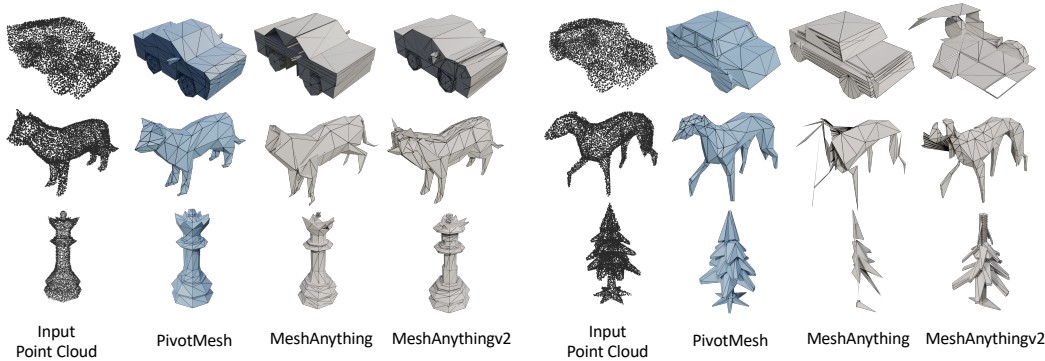

Figure 18: Comparison with concurrent works Meshanything and Meshanythingv2 for point-cloud conditioning.

## A.11 DATA DISTRIBUTION

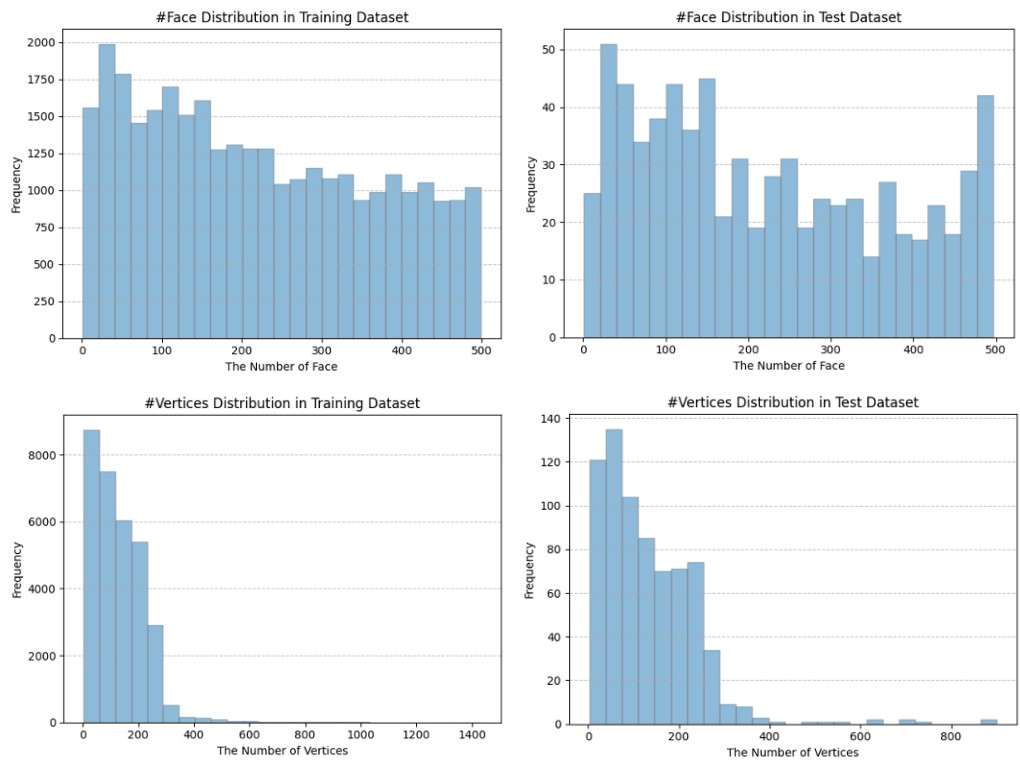

Figure 19: The distribution for the number of vertices and faces in the training and testing data.

