# OpenReview forum: "PivotMesh: Generic 3D Mesh Generation via Pivot Vertices Guidance"
_ICLR.cc/2025/Conference — ICLR 2025 Poster_

### Official Review · Reviewer_EqsP · 2024-10-31

**Soundness:** 3
**Presentation:** 2
**Contribution:** 3
**Rating:** 6
**Confidence:** 3

**Summary:**

This paper focuses on conditional triangle mesh generation based on the pivot vertices of the provided mesh.  The authors proposed to tokenize the mesh with transformer, rather than the GCN architecture used by MeshGPT. The pivot-guided method is shown to outperform PolyGen and MeshGPT.

**Strengths:**

- A transformer-based auto-encoder is proposed to encode the triangle meshes into discrete tokens.
- An auto-regressive transformer is proposed to generate the complete mesh tokens from pivot vertices.
- By conditioning on pivot vertices, the proposed method outperforms PolyGen and MeshGPT.
- Downstream applications of the method are shown for mesh variation and coarse mesh refinement.

**Weaknesses:**

- Both the auto-encoder and auto-regressive transformer use 24 layers of transformers, resulting the method to be memory taken and less efficient.

**Questions:**

- It is confusing how the discrete mesh tokens from auto-encoder and pivot vertices are combined for conditional mesh generation.
- In lines 303-306 of page 6, the authors mentioned about face filtering in the datasets. Is there a distribution map to summarize the number of vertices and faces in the training and test meshes?

---

> ### Author Response · Authors · 2024-11-22
> **Reply to Reviewer EqsP**
>
> We would like to express our gratitude for your thorough review and for raising important points regarding the efficiency and clarity of our methodology.
>
> > Both the auto-encoder and auto-regressive transformer use 24 layers of transformers, resulting the method to be memory taken and less efficient.
>
> Thank you for raising your potential concerns about the efficiency of our model.
> We understand the importance of computational efficiency in practical applications.
> Although our model utilizes 24 layers of transformers, the total number of parameters for both the auto-encoder and auto-regressive transformer is approximately 500M,
> which is considered a acceptable size compared with current LLMs (typically over 7B).
> At inference, our model only takes around 12G VRAM and 1 minute to generate a mesh. We believe this performance is efficient enough for current applications.
>
> > It is confusing how the discrete mesh tokens from auto-encoder and pivot vertices are combined for conditional mesh generation.
>
> We appreciate your feedback on the clarity of conditional mesh generation.
> Similar to recent advanced Multi-modal LLMs, we use prefix conditioning for mesh generation, where pivot vertices and mesh tokens are concatenated a full sequence for Transformer training.
> At inference, the pivot vertices are first extracted from the given meshes and served as the prompts for Transformer.
> Subsequently, mesh tokens are generated conditioned on the prefix pivot vertices.
>
> > In lines 303-306 of page 6, the authors mentioned about face filtering in the datasets. Is there a distribution map to summarize the number of vertices and faces in the training and test meshes?
>
> Thanks for your suggestions on data visualization. We have added the data distribution map for Objaverse on supplementary Figure 19.

---

> > ### Comment · Reviewer_EqsP · 2024-11-25
> >
> > Thanks for the reply and for providing the data distributions. In general, this work contributes to shape generation by enhancing diversity and accuracy, although currently limited to highly decimated meshes. Therefore, I recommend accepting the work.

---

### Official Review · Reviewer_szcv · 2024-11-01

**Soundness:** 4
**Presentation:** 3
**Contribution:** 3
**Rating:** 8
**Confidence:** 5

**Summary:**

In this work, the authors propose a novel framework for explicit mesh generation that can be trained on large-scale datasets. They employ a transformer-based autoencoder to obtain discretized tokens representing the mesh, which are later decoded hierarchically. Additionally, the generative networks are designed to first produce a sequence of pivot mesh vertices as a coarse representation, then use these as conditions to generate the actual mesh tokens. Experimental results demonstrate that the proposed framework achieves comparable and often superior performance to concurrent works. The authors also showcase various downstream applications of their method.

**Strengths:**

- The exploration of generating pivot vertices before faces appears to be a novel approach. Ablation studies validate its effectiveness, particularly without pivot guidance and degrees selection.
- Extensive experiments compare unconditional generation quality across both small and large datasets, such as ShapeNet and ObjaverseXL. The proposed framework outperforms existing methods in all settings.
- The framework demonstrates versatility through various applications, including point cloud-guided generation, mesh variations, and coarse mesh refinement.
- A novel analysis ensures the diversity and originality of the generated objects.

**Weaknesses:**

Although concurrent works are mentioned, several studies have begun exploring the generation of compact meshes in large datasets, demonstrating some success to a certain extent. It is recommended to revise the introduction and abstract to avoid claiming that existing work is difficult to extend to large datasets.

**Questions:**

- Including MeshXL's quantitative results would be highly beneficial, despite its use of a different training dataset. While this is concurrent work and the results would be for reference only, they could serve as a valuable benchmark, helping readers better understand the effectiveness of the proposed method.
- The paper claims that the proposed pivot mesh formulation can generate shorter sequences, providing estimated sequence lengths in Table 1. However, measuring these lengths on a real dataset would offer more clarity and insight into the method's efficiency.

Justification of Rating:
Overall, this work provides a novel direction for compact mesh generation and demonstrates its effectiveness in various aspects. The quantitative and qualitative comparisons show that it achieves good performance compared to existing and concurrent works. Additionally, various applications have outlined its potential beyond unconditional generative tasks. Despite some minor suggestions, I am currently leaning towards acceptance of this work.

**Details Of Ethics Concerns:**

N.A.

---

> ### Author Response · Authors · 2024-11-22
> **Reply to Reviewer szcv**
>
> We are grateful for your insightful feedback, which has been instrumental in refining our manuscript and clarifying our contributions.
> Your suggestions have helped us to improve the presentation of our work.
>
> >  It is recommended to revise the introduction and abstract to avoid claiming that existing work is difficult to extend to large datasets.
>
> Thanks for your valuable suggestions on improving the paper writing and make our contributions clearer.
> We have revised the abstract and introduction to avoid such claims and updated the paper (marked in blue).
>
> > Including MeshXL's quantitative results would be highly beneficial, despite its use of a different training dataset.
>
> We appreciate your suggestion to include the quantitative results of MeshXL, despite its use of a different training dataset.
> We have added the quantitative results of MeshXL for Objaverse on the following table for reference.
>
> |  Methods  |  COV(%)↑  | MMD(10<sup>-3</sup>)↓ | 1-NNA(%)↓ |
> | :-------: | :-------: | :-------------------: | :-------: |
> |  PolyGen  |   23.86   |         24.01         |   84.07   |
> |  MeshGPT  |   35.03   |         17.30         |   63.86   |
> |  MeshXL   |   44.41   |         17.63         |   62.41   |
> | PivotMesh | **46.48** |       **16.66**       | **58.55** |
>
> >  Measuring the mesh sequence lengths on a real dataset would offer more clarity and insight on Table 1.
>
> Thank you for your suggestion to measure the mesh sequence lengths on a real dataset, which would provide more clarity and insight.
> We have computed the average sequence lengths on the Objaverse training dataset for all methods and present the results in the following table.
>
> |     Methods     | MeshGPT | MeshXL | MeshAnything | PivotMesh |
> | :-------------: | :-----: | :----: | :----------: | :-------: |
> | Sequence Length |  1324   |  1986  |     1986     |   1336    |

---

> > ### Comment · Reviewer_szcv · 2024-11-25
> >
> > Thanks for the author's detailed responses. They have clarified my concerns. I will maintain my current rating and recommend acceptance of this work.

---

### Official Review · Reviewer_2uyx · 2024-11-03

**Soundness:** 3
**Presentation:** 3
**Contribution:** 3
**Rating:** 6
**Confidence:** 5

**Summary:**

The paper introduces PivotMesh, a framework for generating compact and detailed 3D meshes at scale. The approach leverages a transformer-based auto-encoder to encode meshes into discrete tokens, which are then decoded hierarchically from face to vertex level. A key contribution is the use of pivot vertices as a coarse representation to guide the generation of complete mesh tokens, allowing for complex topology modeling. The method is evaluated on diverse datasets, including ShapeNet, Objaverse, and Objaverse-xl, demonstrating its versatility and effectiveness in generating high-quality 3D meshes.

**Strengths:**

1. PivotMesh addresses a significant challenge in 3D mesh generation by proposing a scalable framework that can handle large-scale datasets with simplified triangles. The use of pivot vertices as a coarse representation for guiding mesh generation is innovative and effectively handles complex topologies.
2. The paper provides a thorough evaluation of PivotMesh across various datasets and applications, including mesh generation, variation, and refinement. The generated meshes are of high quality, with sharp details and complex geometries, outperforming existing methods.

**Weaknesses:**

1. Diversity of Generated Meshes: While the paper shows diverse mesh generation, a more systematic analysis or quantification of diversity could strengthen the results. The evaluations on more diverse shapes with complex topologies are encouraged to be conducted, such as some shapes with lots of holes, the thin structures (ficus in nerf-synthetic data), and so on.
2. More direct comparisons with the current state-of-the-art methods, especially in terms of computational efficiency and mesh quality, could be beneficial. There is more relative work, such as meshanything(2), EdgeRunner, the comparison is very essential to demonstrate the superiority of the proposed method.
3. The limitations and failure cases should be discussed comprehensively. And the paper acknowledges that the controlling ability of PivotMesh is not sufficient, and sometimes undesired geometries are produced. Enhancing the control mechanisms could be a valuable addition.

**Questions:**

The paper is well-written and presents a contribution to the field of 3D mesh generation. I am impressed with the quality of the generated results, but there are some minor issues: the comparison with other similar baselines and more diverse generated shapes. With the above suggestions addressed, the paper would be a strong candidate for publication. I recommend accepting this paper with minor revisions.
Detailed questions please refer to weaknesses.

---

> ### Author Response · Authors · 2024-11-22
> **Reply to Reviewer 2uyx**
>
> We sincerely appreciate your insightful comments and suggestions, which have been instrumental in enhancing the quality and depth of our paper.
> We will answer your questions as follows:
>
> > The evaluations on more complex topologies (shapes with holes, thin structures) are encouraged to be conducted.
>
> Thanks for your suggestions to show more results in complex topologies.
> We recognize the importance of demonstrating PivotMesh's capabilities in handling such complexities.
> In the supplementary, we add Figure 17 to show the generated meshes with more complex topologies (shapes with holes and thin structures as you mentioned).
> It highlights the robustness of PivotMesh to generate complex topologies.
>
> > Comparison with the concurrent works MeshAnythingv2 and EdgeRunner.
>
> 1. To further show the effectiveness of our method, we compare the results with MeshAnythingv2 as in Figure 18.
>
> 2. Regarding EdgeRunner, we understand the importance of this comparison.
>     However, as its code and checkpoint are not publicly available and re-implementing the algorithm would require an unaffordable amount of resources (training for 7 days on 64 80G-A100 GPUs), we have decided to postpone this comparison to future work.
>
> 3. We compare the computational efficiency with other baselines on the following table. We measure the average number of generated tokens per second on a A800 machine.
>     Although with the similar number of parameters, PivotMesh is slightly slower than other methods, which mainly comes from the different implementation of Transformer.
>     However, our model is able to generate compact meshes with more complex topology and yeilds better performance.
>     We will leverage better implementation of Transformer for PivotMesh in the futhure.
>
> |         Methods         | MeshGPT | MeshXL | MeshAnything | MeshAnythingv2 | PivotMesh |
> | :---------------------: | :-----: | :----: | :----------: | :------------: | :-------: |
> | Inference Speed (tokens/s) |  86.30  | 99.28  |    111.25    |     117.61     |   95.14   |
>
> > The limitations and failure cases should be discussed comprehensively. Enhancing the control mechanisms could be a valuable addition.
>
> Thank you for your suggestion to discuss the limitations and failure cases of PivotMesh comprehensively.
>
> 1. We have included some failure cases in Figure 10 to provide a clearer understanding of PivotMesh's limitations (and we put it on the supplementary due to the page limitation.)
>
> 2. We also believe that it is important to enhance the control mechanisms of PivotMesh.
>     Therefore, we combine fine-grained point-cloud conditions with pivot vertices guidance to further improve the controlling capability.
>     The results with better conditional alignment are shown in Figure 6.

---

> > ### Comment · Reviewer_2uyx · 2024-11-25
> > **response**
> >
> > Thanks for the great efforts in preparing the detailed response. The response has addressed my major concerns and suggested the authors add these revisions into the paper. I still maintain the original scores and lean toward acceptance.

---

### Author Response · Authors · 2024-11-22
**Overall Reply**

Dear Reviewers,

We would like to express our sincere gratitude for your insightful and constructive feedback.
Your expertise and diligence in reviewing our manuscript have been invaluable, and we have taken your comments to heart as we have revised our paper.

In response to your suggestions, we have made the following revisions to the paper (**marked in blue**):

- We add Figure 17 to show the generated meshes with more complex topologies with holes and thin structures. (Reviewer 2uyx)
- We add Figure 18 to compare with the concurrent work MeshAnythingv2 and an additional table to compare the model efficiency. (Reviewer 2uyx)
- We revised the abstract and introduction to avoid claiming that existing works are difficult to extend to large datasets. (Reviewer szcv)
- We add two tables to show the quantitative results of MeshXL and the real sequence length on Objaverse. (Reviewer szcv)
- We add Figure 19 to show the data distribution map for Objaverse. (Reviewer EqsP)

---

### Meta-Review · Area_Chair_C3Sp · 2024-12-18

**Metareview:**

This paper introduces a novel generative model for 3D meshes designed to train on a large-scale dataset. To effectively train on such a dataset, the authors first generate a sequence of pivot meshes, a coarser representation of the mesh, and then use these as a condition to generate the actual fine mesh. The authors also employ a transformer-based hierarchical autoencoder to represent a mesh with a sequence of discretized tokens. The experimental results demonstrate the superior performance of the proposed method compared to concurrent work and the effectiveness of generating pivot vertices as an intermediate representation.

All reviewers agreed that the idea of using pivot vertices is novel and innovative, and its effectiveness has been demonstrated through the experimental results. The versatility of the proposed method in point-cloud-guided generation, mesh variations, and coarse mesh refinement was also appreciated. The discussion after the rebuttal quickly converged to acceptance.

**Additional Comments On Reviewer Discussion:**

Please see the Metareview.

---

### Decision · Program_Chairs · 2025-01-22

Accept (Poster)